# Hybrid Finite-Discrete Element Modeling of the Mode I Tensile Response of an Alumina Ceramic

**Jie Zheng \*** , **Haoyang Li** and **James D. Hogan**

Department of Mechanical Engineering, University of Alberta, Edmonton, AB T6G 2R3, Canada
\* Correspondence: jzheng11@ualberta.ca

**Abstract:** We have developed a three-dimensional hybrid finite-discrete element model to investigate the mode I tensile opening failure of alumina ceramic. This model implicitly considers the flaw system in the material and explicitly shows the macroscopic failure patterns. A single main crack perpendicular to the loading direction is observed during the tensile loading simulation. Some fragments appear near the crack surfaces due to crack branching. The tensile strength obtained by our model is consistent with the experimental results from the literature. Once validated with the literature, the influences of the distribution of the flaw system on the tensile strength and elastic modulus are explored. The simulation results show that the material with more uniform flaw sizes and fewer big flaws has stronger tensile strength and higher elastic modulus.

**Keywords:** hybrid finite-discrete element model; alumina ceramic; crack; tension; flaw

## 1. Introduction

Advanced ceramics are often used as structural components in shielding applications [1–4] because of their desirable properties, such as low density [5], high hardness [6], and high wear resistance [7]. In ballistic applications, the role of the ceramic upon impact is to blunt the projectile [8–10] and to distribute the load [9]. During impact, pre-existing micro-cracks or pores in ceramics serve as stress concentration sites [11], which significantly affect fracture behavior [12–15]. The relationships between microstructure and failure processes have been widely investigated [14,16–20]. For example, Munro [16] and Nohut [17] found that the strength of alumina ceramics is limited by the distribution of flaws in the material, and the Weibull analysis could be used to characterize both the strength and the flaw system. Recently, Lo et al. [14] studied the microstructural and mechanical variability of AD85 alumina, and they found the flaw characterization (e.g., pore size, spatial distribution, orientation, and morphology) significantly influences the mechanical response of the alumina ceramic. In a separate study, Hogan et al. [18–20] identified the relationship between microstructure and fragment size by observing impact and compression experiments of ceramics. To better understand the failure mechanisms of advanced ceramics, controlled experiments such as uniaxial compression, beam bending tests, and Brazilian disk experiments are often coupled with advanced high-speed imaging [21–29]. In these experiments, the mode I tensile opening cracks are widely accepted as an important failure mode of advanced ceramics [30–34]. However, direct tension tests is a challenge for advanced ceramics because of their brittleness. In the current study, the mode I tensile behavior of an alumina ceramic is investigated. The widespread use of high-purity alumina ($Al_2O_3$) as body armor material is due to its beneficial combination of favorable ballistic properties, affordability, and well-established manufacturing processes [35]. A significant number of studies have been conducted with the objective of enhancing the performance of alumina ceramics [35–38]. These ceramics are usually produced through various sintering techniques such as flash sintering (FS) [38], hot-pressing sintering (HPS) [36], and spark plasma sintering (SPS) [35]. The trend of 3D printing ceramics is gradually becoming

more prevalent, offering multiple methods for production [35]. Some of the most common methods include stereo lithography, which involves curing photo-curable binder loaded ceramic pastes, and selective laser sintering, involving laser sintering of green powder beds [35]. Other techniques such as modified inkjet printing and binder jetting are also utilized in ceramic material production [37]. These different manufacturing methods would affect alumina ceramic's microstructure (e.g., flaws). However, these experimental studies are limited by manufacturing technologies and testing methods for the tensile response of ceramics. For example, many factors (e.g., additives [38], debinding step [36], and temperature [35]) can affect the internal flaws and mechanical performance of ceramics in sintering manufacturing, which is complex [37]. In addition, exploring the influence of flaw systems on direct tensile performance is expensive and challenging [37,39]. To overcome these challenges, in the current study, the inherent microstructural flaws based on experimental studies [16,17,40] are incorporated to the numerical modelling method to more realistically explore the mode I tensile behavior of an alumina ceramic.

Numerous numerical models have been established to describe the failure behavior of brittle materials. These include the continuum damage mechanics (CDM) [41,42], the extended Finite Element Method (XFEM) [43], virtual crack closure technique (VCCT) [44–46], and the cohesive zone method (CZM) [47]. The CDM method utilizes damage parameters to explain the failure process but cannot capture crack-induced discontinuities [48]. VCCT, on the other hand, can simulate pre-defined crack propagation by imposing constraints on the nodes at crack edges but requires re-meshing [44–46]. XFEM avoids mesh refinement and reconstruction by modifying the displacement approximation function in conventional FEM with an enrichment function term [43]. While these methods can model progressive cracking behavior, they necessitate additional limitations, such as external criteria for discontinuous displacement enrichment [43], re-meshing requirements [44–46], and complex model pre-definitions [44–46]. Additionally, these methods cannot effectively address complex cracking problems, including crack intersection, coalescence, and branching. The CZM has several advantages over the other methods, including (1) creating new surfaces to generate cracks, (2) allowing for branched and intersecting cracks, and (3) eliminating the singularity present in linear elastic fracture mechanics [49]. In the literature [50,51], the CZM framework has been applied to study the strength of advanced ceramics, dynamic fracture events [52], and fragmentation of brittle materials [53]. In the CZM method, new surfaces are created when the fracture occurs, and these new faces require numerical contact algorithms, which makes CZM ideally suited for discretized methods [54]. To address the mutually interacting separate fragments in fracture processes, Munjiza [48] developed an innovative numerical approach called the hybrid finite-discrete element method (HFDEM). One distinct feature of the HFDEM is that it is able to capture the transition from a continuum (e.g., finite element method) to a discontinuous-based method (e.g., discrete element method) [55,56] to overcome the inability of these methods to capture progressive damage and failure processes in brittle materials (e.g., geomaterials [57–59] and ceramics [56]). In HFDEM, materials are often discretized as triangle elements (two-dimensional) or tetrahedral elements (three-dimensional), and cohesive elements are utilized to connect these discrete elements to represent the potential arbitrary crack path [60]. In addition, the use of tetrahedral elements offers the advantage of generating more potential fracture surfaces when compared with hexahedron elements, which make the results using tetrahedral element more reliable [48,60–63]. Then, an explicit finite difference time integration scheme is applied to solve the motion of the discretized system [48]. Recently, HFDEM has been used in modelling brittle materials under different loading conditions, as it can explicitly describe the process of fracture nucleation and growth, as well as the interaction of newly created discrete fragments [54,60,64]. In the current study, the HFDEM is applied to investigate the failure process of the alumina ceramic.

Motivated by these previous studies, this paper aims to develop a three-dimensional HFDEM model to investigate the mode I tensile opening failure of the alumina ceramic. This method is focused on the macroscopic failure process accounting for the flaw system

with an implicit method. The material is assumed to have Weibull distributed initial flaws, making its failure mode stochastic. The failure of the specimen is simulated through the nucleation and propagation of these cracks generated by flaws. Limited experimental results are available for the direct tension problems, owing to the challenges posed by testing, such as specimen gripping and alignment [65]. The indirect tensile strength from our previous study is used [29], and the simulation results show reasonable agreement with the experimental results. The influence of flaw system on the tensile strength is also investigated. Overall, the current study provides new insight into the mode I tensile fracture behavior of alumina ceramic.

## 2. Computational Approach

In this section, a three-dimensional hybrid finite-discrete element method (HFDEM) is established to describe the failure process of brittle materials, and then applied to the alumina ceramic. First, we develop the main features of the cohesive law to represent the mode I tensile response of brittle materials. Second, the distribution of the flaw system [16,17,40] is considered in this model. Finally, the model is implemented with a FORTRAN vectorized user-material (VUMAT) subroutine in ABAQUS/Explicit to solve the model numerically.

### 2.1. The Cohesive Law

An extensive account of the HFDEM theories and their finite element implementation can be found in [48,55–59,66]. In this section, we summarize the main features of the cohesive law used in the current study:

$$\sigma = (1 - D)K\delta \tag{1}$$

with

$$\sigma = \begin{bmatrix} \sigma_1 \\ \tau_1 \\ \tau_2 \end{bmatrix}, \quad K = \begin{bmatrix} K_1 & 0 & 0 \\ 0 & K_2 & 0 \\ 0 & 0 & K_3 \end{bmatrix}, \quad \delta = \begin{bmatrix} \delta_1 \\ \delta_2 \\ \delta_3 \end{bmatrix}$$

where $\sigma$ represents the interface stress, $\sigma_1$ is the normal stress, and $\tau_1$ and $\tau_2$ are the shear stresses in the other two directions. $D$ is a scalar damage parameter of the REA, where $D = 0$ is the intact state, and $D = 1$ represents a fully damaged state. $K$ is the penalty stiffness of the interface where subscript 1 represents the normal direction, and subscripts 2 and 3 represent the two shear directions. $\delta$ represents relative displacements.

The linear irreversible cohesive law is widely used for the decaying response of brittle materials, such as rocks [54,60,64] and ceramics [51,66–68]. For a damage value in the range of 0~1, the damage evolution can be expressed as:

$$D = \max\left\{0, \min\left\{1 - \left(\frac{\delta_c - \delta_e}{\delta_c - \delta_m^0}\right), 1\right\}\right\} \tag{2}$$

where $\delta_e$ is the effective relative displacement ($\delta_e = \sqrt{\delta_1^2 + \delta_2^2 + \delta_3^2}$), and $\delta_c$ represents the critical displacement when interface failure occurs. $\delta_m^0$ is the relative displacement when the damage initiates under mixed mode loading, which is obtained by:

$$\delta_m^0 = \sqrt{\frac{\left(\delta_1^0\right)^2 (1 + \beta^2)\delta_2^0\delta_3^0}{\delta_2^0\delta_3^0 + \beta^2\left(\delta_1^0\right)^2}} \quad \text{and} \quad \beta = \frac{\sqrt{\delta_2^2 + \delta_3^2}}{\delta_1} \tag{3}$$

$$\delta_1^0 = \frac{\sigma_1^0}{K_1}, \quad \delta_2^0 = \delta_3^0 = \frac{\tau_1^0}{K_2} = \frac{\tau_2^0}{K_3} \tag{4}$$

where $\sigma_1^0$, $\tau_1^0$, and $\tau_2^0$ are the strengths in the normal and two shear directions under pure mode loading, and $\delta_1^0$, $\delta_2^0$ and $\delta_3^0$ are the corresponding displacements.

Until here, the only unknown parameter in Equation (2) is the critical displacement $\delta_c$. In the cohesive zone method, mode II and mode III fracture is often regarded as the same due to a lack of mode III mechanical property information [69]. Thus, the mixed mode energy-based failure function [70] can be expressed as:

$$\left(\frac{G_1}{G_1^c}\right)^\gamma + \left(\frac{G_{shear}}{G_{shear}^c}\right)^\gamma = 1 \tag{5}$$

where $G_{shear}^c$ is the critical shearing energy release rate with $G_{shear}^c = G_2^c = G_3^c$, and $G_1^c$, $G_2^c$ and $G_3^c$ terms are the fracture energies under pure mode loading. The $G_{shear}$ is the energy dissipation rate by shearing, which is sum of the energy release by the mixed mode II and III crack, $G_{shear} = G_2 + G_3$, and the $G_2$ and $G_3$ are given in Equation (6). A quadratic $\gamma = 2$ failure criteria is frequently chosen according to the mixed mode experimental results [70,71] and so it is used here. For the cohesive interface, the energy dissipation rates are:

$$G_i = \int \sigma_i d\delta_i \quad (i = 1, 2, 3) \tag{6}$$

In HFDEM model proposed in this current study, the bonding stresses transferred by the material are functions of the relative displacements across the crack elements, and this is illustrated in Figure 1 for the mode I tensile response.

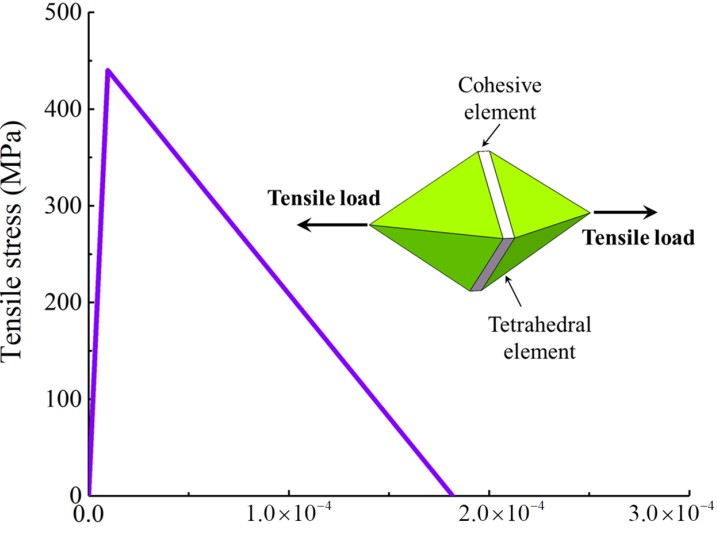

**Figure 1.** The mode I constitutive behavior of the cohesive element with $K_1 = 4.6 \times 10^7$ N/mm$^3$, $\sigma_0 = 440$ MPa, and $G_1^c = 0.04$ N/mm. In the inset figure, a cohesive (crack) element is interspersed throughout two tetrahedral elements.

### 2.2. The Microscopic Stochastic Fracture Model

For brittle materials such as ceramics, the strength is limited by the distribution of flaws in the material specimen, and any flaw in the material can serve as an origin of a crack [16]. The Weibull strength distribution is widely used in ceramics to characterize the influence of flaws statistically [16,17,40].

$$P(\sigma, V) = 1 - \exp\left[-\frac{V}{V_0}\left(\frac{\sigma}{\sigma_0}\right)^m\right] \tag{7}$$

where $P(\sigma, V)$ is the cumulative failure probability of an alumina ceramic, $V$ is the volume of the investigated component, $V_0$ is the characteristic volume, $\sigma$ is the applied stress, $m$ is the Weibull modulus, and $\sigma_0$ is the Weibull characteristic strength.

In HDFEM, the cohesive elements represent the intrinsic and extrinsic flaws [60]. Thus, Weibull's statistical strength theory [17] is applied to cohesive elements to show the stochastic properties of the alumina ceramics. In the recent study by Daphalapurkar et al. [68], a modified microscopic facet-strength probability distribution based on Equation (7) is applied to the dynamically introduced cohesive method.

$$f(\sigma) = \frac{m_0}{\sigma_0} \left( \frac{A}{A_0} \right)^{\frac{m_0}{m_a}} \left( \frac{\sigma}{\sigma_0} \right)^{m_0 - 1} \exp\left[ \left( \frac{A}{A_0} \right)^{\frac{m_0}{m_a}} \cdot \left( \frac{\sigma}{\sigma_0} \right)^{m_0} \right] \tag{8}$$

where $A$ is the facet area of the cohesive element, $A_0$ is the characteristic area; $m_0$ is the Weibull modulus of strength distribution, and $m_a$ is the Weibull modulus for the effective area modification. In the current study, we applied the microscopic facet-strength probability model to the pre-inserted cohesive method to represent the flaws in the material. Monte Carlo simulations generate the strength data affected by flaws according to Equation (8).

Figure 2 shows a Monte Carlo simulation for Weibull's statistical strength distribution of cohesive elements with $m_0 = 11.0$, $m_a = -11.0$, $\sigma_0 = 440.0$, $A = 0.01268$ and $A_0 = 0.013$. The percentage of low-strength cohesive elements (below 350 MPa) is around 7.9%, which is associated with the big flaws in the material. The rest of the cohesive elements (around 92.1 %) have strong strength (between 350 and 530 MPa), which corresponds to the material with smaller flaws. The Weibull statistical model for failure has been adapted to fit within the cohesive element framework by incorporating the following assumptions: (1) The pre-inserted cohesive elements in the finite element mesh are treated as potential locations for flaws. (2) When the tensile stress applied on a facet exceeds its strength, the flaw in that location becomes a microcrack. The cohesive elements with low strength will activate at nearly negligible loads and can be considered as intrinsic microcracks.

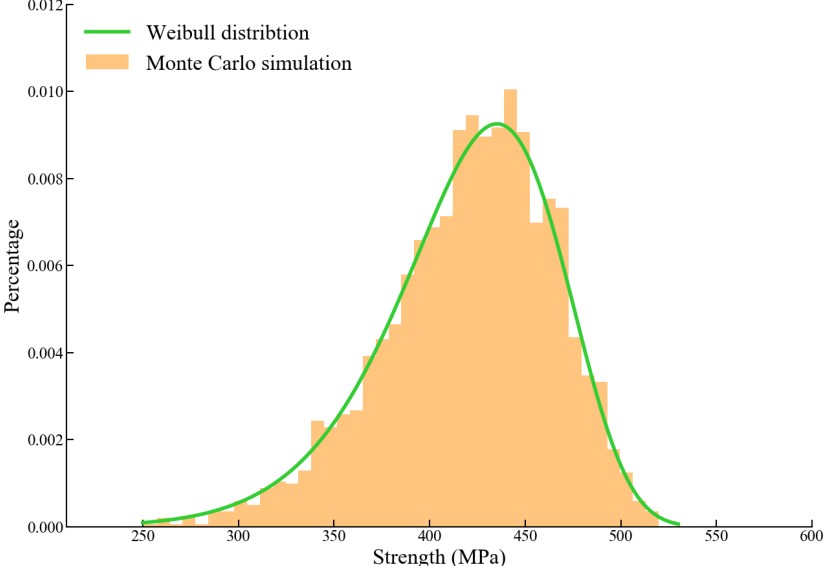

**Figure 2.** The green line is Weibull's statistical strength distribution of cohesive elements obtained from Equation (8) with $m_0 = 11.0$, $m_a = -11.0$, $\sigma_0 = 440.0$, $A = 0.01268$ and $A_0 = 0.013$. The orange bar is the statistics of the facet strength of the cohesive element with random flaws generated by Monte Carlo simulations. The percentage of low-strength cohesive elements (below 350 MPa) is around 7.9%, which is associated with the big flaws in the material. The rest of the cohesive elements (around 92.1%) have strong strength (between 350 and 530 MPa), which corresponds to the material with smaller flaws.

### 3. The Hybrid Finite-Discrete Element Method

The hybrid finite-discrete element method is an advanced numerical method that combines continuum mechanics methods with the discrete element method (DEM) algorithms to solve complicated crack problems involving multiple interacting deformable bodies [48,55,56]. In HFDEM, the specimens are considered a collection of elastic bulk elements connected by cohesive elements [48]. The cohesive elements represent the inherent flaws in the specimens, which become the potential cracks during the loading process, and are also referred to as "crack elements" in HFDEM [60]. When $D=1$, the crack element is completely broken, and the cohesive element is deleted from the model, which generates new crack surfaces. In the HFDEM model, cohesive elements introduce a well-defined length scale into the material description, and are consequently sensitive to the size of the element [72]. In the current study, the physical and statistical model (Equation (8)) has introduced an internal microstructural length scale ($A_0$) into our HFDEM model, and thus regularizes the problem in terms of mesh convergence. This technique (Equation (8)) has been applied to the dynamically introduced cohesive method and demonstrated the effectiveness of mesh convergence [67,68]. However, this technique (Equation (8)) has not been used in the pre-inserted cohesive method before, and we use the pre-inserted cohesive method considering the physical and statistical model (Equation (8)) to simulate the mode I tensile behavior in this paper.

*3.1. Modeling Mode I Failure Considering Distributed Flaws*

In this study, the mechanical properties of the CeramTec 98% are provided by the manufacturer [73] and evaluated in our previous studies [29,74,75]. This CeramTech ceramic has an alumina content of 98 mass percentage, a low porosity of less than 2%, a high hardness of 13.5 GPa, a low density of 3.8 g/cm$^3$, Young's modulus of $E = 335$ GPa, and Poisson's ratio of $v = 0.23$. For the physical and statistical model [16,17,29,40,74,75], the Weibull modulus of the strength distribution is $m_0 = 11$, the Weibull modulus for the effective area modification is $m_a = -11$, the Weibull characteristic strength is $\sigma_0 = 440$ MPa, the characteristic area is $A_0 = 0.013$ mm$^2$, and mode I fracture energy is $G_c = 0.04$ N/mm, which are summarized in Table 1.

**Table 1.** The properties of CeramTec 98% alumina for the HFDEM model.

| The Mechanical Properties of the CeramTec 98% Alumina | |
|---|---|
| Porosity | <2% |
| Hardness | 13.5 (GPa) |
| Density | 3.8 (g/cm$^3$) |
| Young's modulus | 335 (GPa) |
| Poisson's ratio | 0.23 |
| **The Properties for the Microscopic Stochastic Fracture Model** | |
| Weibull modulus of the strength distribution | 11 |
| Weibull modulus for the effective area modification | −11 |
| Weibull characteristic strength | 440 (MPa) |
| Characteristic area | 0.013 (mm$^2$) |
| Mode I fracture energy | 0.04 (N/mm) |

The simulation sample for the mesh sensitivity analysis is a ceramic block with dimensions $L_x = 1.5$ mm, $L_y = 0.5$ mm, and $L_z = 2.5$ mm. In the simulations, a fixed-displacement boundary condition is implemented to the bottom, and the specimen is free to expand or contract freely in the lateral direction. The loading process is performed by imposing a velocity boundary condition in the z-direction to mimic the direct tension loading. The boundary velocity, $v_0$, is fixed to 1 mm/s, which corresponds to quasi-static direct tension loading. This model has also been also employed by Zhou and Molinari [67] to study mesh sensitivity. In the literature [61,72], various studies have focused on the

influence of mesh size on the mechanical response of brittle materials, and the upper limit of mesh size for advanced ceramics is suggested to be around 0.3 mm. In addition, experimentally determined fragment sizes for ceramics [18,19] can also guide in choosing the element size because each tetrahedral elastic element acts as a potential fragment generated in the post-fracture process in the HFDEM model. In their experimental study, Hogan et al. [18] measured more than 1500 ceramic fragments generated by quasi-static uniaxial compression. According to their measurements, most of the fragments are between 0.01 and 1 mm in size, and over 70% of the fragments are larger than 0.1 mm [19]. With this taken into account, the current study focuses on capturing the macroscopic failure (fracture and fragmentation process) numerically, therefore, we performed simulations with mesh sizes of 0.075, 0.1, 0.15, and 0.25 mm to confirm that there was minimal sensitivity. Monte Carlo simulations are carried out to simulate the variety of the specimen strength because of the different flaw distributions inherent to the samples. Three numerical tests are performed for each simulation case (fixed mesh and material parameters). The facet area of the cohesive element is $A = 0.00336$ mm$^2$ for mesh sizes of 0.075 mm, $A = 0.00611$ mm$^2$ for mesh sizes of 0.1 mm, $A = 0.01268$ mm$^2$ for mesh sizes of 0.15 mm, and $A = 0.038$ mm$^2$ for mesh sizes of 0.25 mm, which are summarized in Table 2.

**Table 2.** Statistics of four FEM meshes.

| Validation Models | Average Mesh Size (mm) | Average Facet Area (mm$^2$) |
|---|---|---|
| Case 1 | 0.25 | 0.038 |
| Case 2 | 0.15 | 0.01268 |
| Case 3 | 0.1 | 0.00611 |
| Case 4 | 0.075 | 0.00336 |

The stable time step used in the current study satisfies the criteria given by [68]:

$$\Delta t_{stable} \leq \alpha \left( \frac{l_e}{c} \right) \tag{9}$$

where $c$ is the wave speed in the alumina ceramic, $l_e$ is the smallest element size, and $\alpha$ is a factor whose value is 0.1 or less. This stable time step is able to resolve the two kinds of time-scales. The first time-scale is the response time associated with the cohesive law, and is given by [76]:

$$t_0 = \frac{E}{c} \frac{G_c}{\sigma_0{}^2} \tag{10}$$

The other time-scale is associated with the time required for complete decohesion of the cohesive law [68]:

$$t_c = \left( \frac{\delta_c}{c\dot{\varepsilon}} \right)^{0.5} \tag{11}$$

where $\dot{\varepsilon}$ is the applied strain rate, and $\delta_c$ is the critical crack opening displacement.

Figure 3a illustrates the tensile stress and strain evolution of the alumina under quasi-static direct tension loading. The stress increases linearly during the loading process with a constant strain rate until catastrophic failure occurs. The variety of the tensile strength in the simulations with the same mesh size is due to the different flaws in the materials generated by Monte Carlo simulations. The peak stresses of the four kinds of simulation with different mesh sizes are consistent with the experimental results [29], and the difference of simulated stress-strain responses are within 7% (see Figure 3b). Although the difference in stress (or strain) states between indirect and direct tension samples, the difficulties inherent in conducting conventional direct tensile tests on advanced ceramics have resulted in using indirect methods, such as Brazilian tests, for evaluating their tensile strength [23,27,29]. Our previous investigation [29] involved the application of an experimental approach coupled

with a theoretical method to determine the tensile strength of alumina ceramic materials, which makes the obtained tensile strength more reliable.

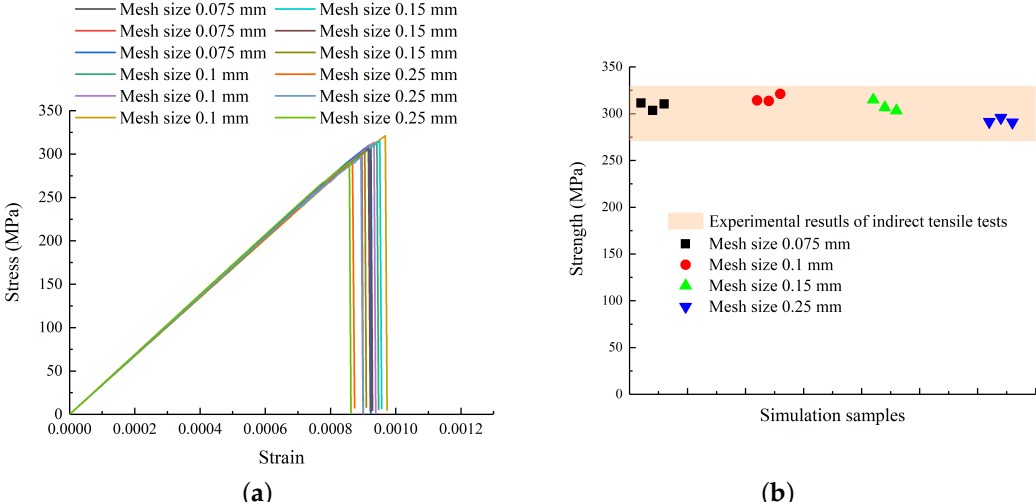

**Figure 3.** (**a**) Variation of engineering stress-strain response of the CeramTec 98% alumina during the direct tension simulations with four kinds of mesh sizes. (**b**) The shaded region is the tensile strength of the CeramTec 98% obtained by Brazilian disk experiments [29], and the dots are the simulation results with four kinds of mesh sizes

Next, the failure pattern of the simulation with four different mesh sizes is shown in Figure 4. A horizontal crack perpendicular to the loading direction is observed during the loading process. The single main crack causes catastrophic failure at the low loading rate. Some fragments appear near the crack surfaces due to crack branching. The fracture process of simulation results under the tensile loading is consistent with the observation in brittle materials [77–80]. The origins of fractures can stem from either internal volume flaws (such as cracks, pores, uneven density, and composition variations) or surface flaws (like cracks from machining, surface pits, and voids) [80]. In the current study, we consider the tensile characteristics in the loading direction based on the statistical studies of flaws for alumina ceramics [16,17,40]. However, the flaws on the sample surfaces and the influence of the flaws on shear or compression directions are not considered. In brittle materials, fracture in brittle materials occurs due to the application of stress to the critical flaws in the material leading to unstable propagation of that cracks. When subjected to direct tension testing, the crack at the origin expands approximately perpendicular to the principal tensile stress and spreads symmetrically from the origin in a uniform stress field. The fracture processes include crack surface creation, fragment release, and the main crack splits into multiple branches [80]. In the simulation, the branching phenomenon is more pronounced for the samples with smaller mesh sizes, and this is because each cohesive element associated with flaws is a potential source of microcracking in the material. The smaller meshed sample has more cohesive elements (or crack elements) with critical strength in the material. The increase in the number of crack initiation sources naturally promotes branching behavior [77–80]. Overall, our model is mesh-independent based on qualitative (e.g., failure patterns) and quantitative (e.g., stress-strain curves) evaluations.

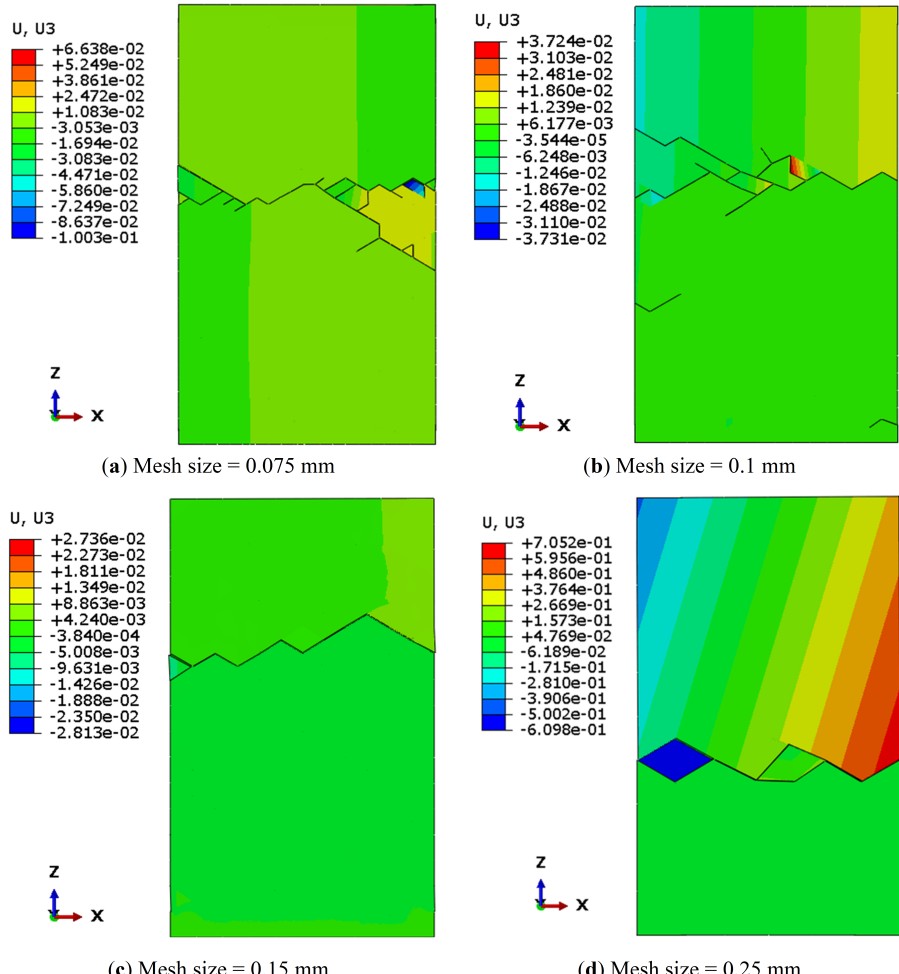

(**a**) Mesh size = 0.075 mm                                  (**b**) Mesh size = 0.1 mm

(**c**) Mesh size = 0.15 mm                                  (**d**) Mesh size = 0.25 mm

**Figure 4.** The failure pattern of the samples with different mesh sizes. The legends in the figure correspond to the displacement of the samples in the z-direction ($U_3$). It is observed that a horizontal crack perpendicular to the loading direction is generated during the loading process. The single main crack causes catastrophic failure. In some cases, the main crack may split into multiple branches. Some fragments appear near the crack surfaces due to crack branching.

### 3.2. The Effect of Flaw Distribution

In recent years, the influences of cohesive parameters (i.e., cohesive strength and the critical energy release rate) on the mechanical response of brittle materials have been widely investigated [54,60,62,64]. However, limited numerical studies focused on the influence of the distribution of the flaw systems. The influence of the flaw systems in materials on the strength at the quasi-static loading rate has traditionally been related to the size of the largest flaw in the material based on experimental studies [68,81]. Moreover, researchers found that the distribution of the flaw systems is also significant to the elasticity and strength of ceramic materials [82,83]. In the current study, we consider the different Weibull distributions of the flaw system with five choices of Weibull modulus $m_0$ = 9, 10, 11, 12, and 13 [16]. Figure 5 shows that the percentage of low-strength cohesive elements (below 350 MPa) is around 12.2% for $m_0$ = 9, 9.9% for $m_0$ = 10, 7.9% for $m_0$ =11, 6.4% for $m_0$ = 12, and 5.1% for $m_0$ = 13. This means the higher value of $m_0$ corresponds to more uniform flaw sizes with fewer big flaws, and a smaller $m_0$ physically represents a more heterogeneous material with a higher amount of bigger flaws.

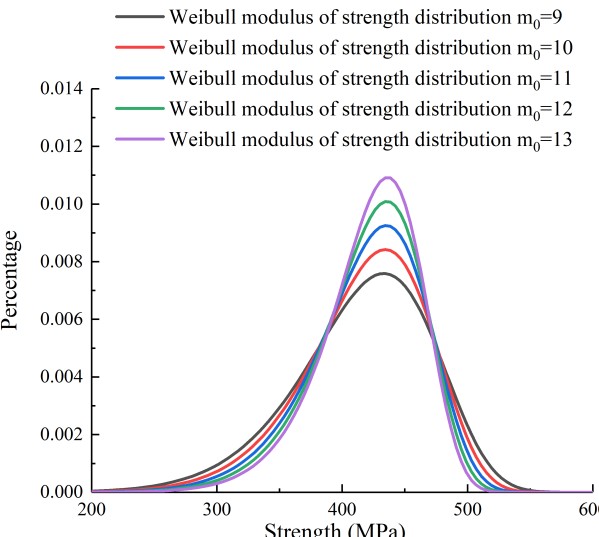

**Figure 5.** Microscopic strength distributions with different Weibull modulus ($m_0$ = 9, 10, 11, 12, and 13).

Figure 6a shows the tensile stress-strain curves during the constant low strain-rate loading, and Figure 6b summarizes the tensile strength and elastic modulus obtained by simulation with five different Weibull modulus. It is observed that material with smaller $m_0$ (i.e., more heterogeneous material with a higher number of bigger flaws) exhibits lower tensile strength and elastic modulus. This agrees with the experimental observation [82,84] that both the strength limit and elasticity moduli decrease with a higher percentage of big flaws. This is because the strength of a ceramic specimen is not only determined by the existing critical flaws in the material [85] but also by the flaw distribution [16,17,40]. Next, Figure 7 shows that the failure pattern is consistent with Figure 4, for which a single main crack perpendicular to the loading direction is generated during the loading process. Some fragments appear near the crack surfaces due to crack branching. The branching phenomenon is more pronounced for the material with a smaller $m_0$ value. This is because the material has more critical flaws with larger sizes, and these flaws are activated during the loading process. The increase in the number of crack initiation sources naturally stimulates the branching behavior.

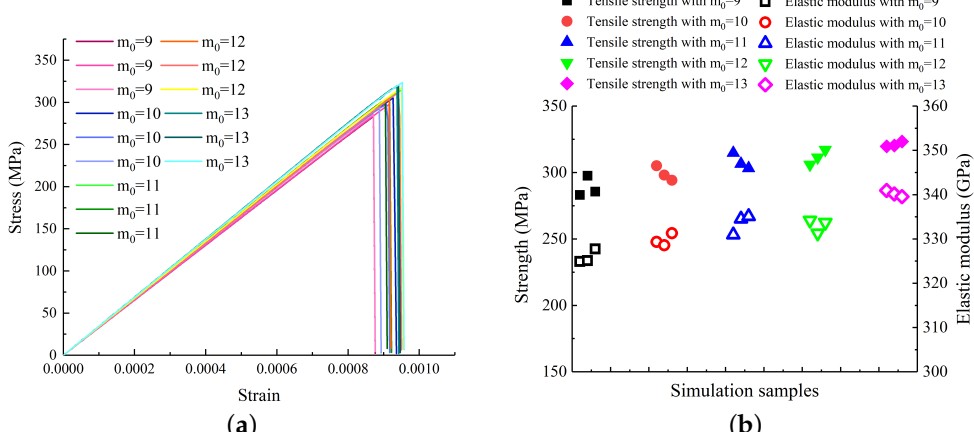

**Figure 6.** (**a**) The engineering stress-strain response of the CeramTec 98% alumina during the direct tension simulations with different Weibull modulus ($m_0$ = 9, 10, 11, 12, and 13). (**b**) The full dots are the tensile strength and the hollow dots are the elastic modulus obtained by simulation with five different Weibull modulus.

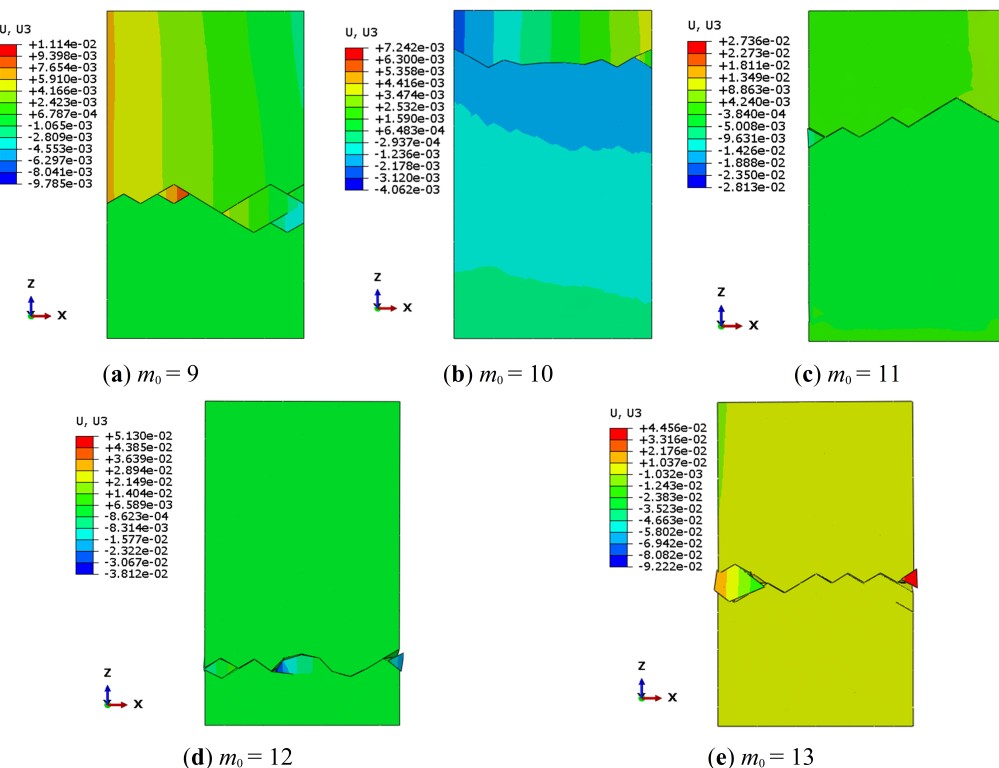

**Figure 7.** The failure pattern of the simulation results with different Weibull modulus ($m_0 = 9, 10, 11,$ 12, and 13). The $U_3$ legend in the figure indicates the displacement of the samples in the z-direction. The failure pattern is consistent with Figure 4, a horizontal crack forms perpendicularly to the loading direction during the loading process, ultimately leading to a catastrophic failure caused by the single main crack. In certain instances, the main crack can divide into multiple branches, resulting in fragments near the crack surfaces due to crack branching.

## 4. Conclusions

We performed a three-dimensional HFDEM model to investigate the mode I tensile opening failure of the alumina ceramic. In this model, the ceramic material was divided into two parts: bulk material regions, represented by tetrahedral elements, and pre-inserted cohesive elements that appear at the interfaces (facets) between the tetrahedral elements. The bulk material was linear, homogeneous, and isotropically elasticit, while the behavior of the micro-cracks was described using cohesive elements. A microscopic stochastic fracture model is developed considering a random distribution of internal flaws in the material. The microscopic stochastic fracture model, including a Weibull strength distribution, is adapted to fit within the HFDEM model. Our model implicitly considered the tensile failure processes related to the flaw system in the material and explicitly showed the macroscopic failure patterns. This model is mesh-independent based on qualitative (e.g., failure patterns) and quantitative (e.g., stress-strain curves) evaluations. The tensile strength obtained by our model is consistent with the indirect tensile testing results from our previous study [29]. In the simulation, micro-cracks nucleated randomly within the sample and grew, eventually merging, leading to the catastrophic failure of the specimen. A single main crack perpendicular to the loading direction is observed during the tensile loading process. Some fragments appear near the crack surfaces due to crack branching. Furthermore, the influences of the flaw system distribution on the tensile strength and elastic modulus are explored. The simulation results show that the material with more uniform flaw sizes and fewer big flaws has stronger tensile strength and higher elastic modulus. Overall, applying this new model provides theoretical guidance for future material design and optimization.

**Author Contributions:** Conceptualization, J.Z., H.L. and J.D.H.; methodology, J.Z.; software, J.Z.; validation, J.Z., H.L. and J.D.H.; formal analysis, J.Z.; investigation, J.Z.; resources, J.D.H.; writing—original draft preparation, J.Z.; writing—review and editing, H.L. and J.D.H.; visualization, J.Z.; supervision, J.D.H.; project administration, H.L.; funding acquisition, J.D.H. All authors have read and agreed to the published version of the manuscript.

**Funding:** This research was funded by Defence Research and Development Canada, grant number NSERC ALLRP 560447-2020. This research was funded by General Dynamics Land Systems–Canada, grant number NSERC ALLRP 560447-2020. This research was funded by NP Aerospace, grant number NSERC ALLRP 560447-2020. The funds receiver is James D. Hogan.

**Data Availability Statement:** Data will be made available on request.

**Acknowledgments:** This work is supported by Defence Research and Development Canada (DRDC), General Dynamics Land Systems–Canada, and NP Aerospace through NSERC Alliance project ALLRP 560447-2020. The views and conclusions contained in this document are those of the authors and should not be interpreted as representing the official policies, either expressed or implied, of General Dynamics, NP Aerospace, DRDC or the Government of Canada. The Government of Canada is authorized to reproduce and distribute reprints for Government purposes notwithstanding any copyright notation herein.

**Conflicts of Interest:** The authors declare the following financial interests/personal relationships which may be considered as potential competing interests: James D. Hogan reports financial support was provided by Defence Research and Development Canada. James D. Hogan reports financial support was provided by General Dynamics Land Systems Canada. James D. Hogan reports financial support was provided by NP Aerospace.

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
