# Peer review of "Hybrid Finite-Discrete Element Modeling of the Mode I Tensile Response of an Alumina Ceramic"

_2673-3951, doi:10.3390/modelling4010007_

Round 1

Reviewer 1 Report

Mode-I fracture behavior of a brittle material such as alumina ceramic is simulated using a three-dimensional hybrid finite-discrete element model. The crack surface is idealized by a cohesive element with variable stiffness with respect to deformation. Three-dimensional fracture formulation and mesh sensitivity analysis are quite impressive in the study. The authors need to clarify the following points.

 1) Authors need to highlight the novelty of their work compared to previous contributions from the literature.

2) It seems that the micro-level cracks are assumed normal to the load direction. But in a porous medium or case of any manufacturing defect the flaws in general, might be inclined. Authors need to specify the limitations of their methodology.

3) The virtual crack closure technique (VCCT) was established for fracture studies. The research study will be more interesting if the present technique is compared with VCCT. Authors are advised to include short literature on VCCT and highlight their methodology over VCCT.

Reviewer 2 Report

In the paper "Hybrid finite-discrete element modeling of the mode I tensile response of an alumina ceramic" a three-dimensional hybrid finite-discrete element model has been developed to investigate the mode I tensile opening failure of the alumina ceramic.

In general, the work is interesting and actual. However, I would like to clarify some points:

1. Please explain what is displayed in Figures 4 and 7, what is denoted as U and U3?

2. Why are the axes in Figures 4(e) and 7(b) different from those in other Figures 4 and 7?

3. Will the boundary conditions affect the calculation results, why is the boundary velocity of 1 mm/s chosen?

4. The paper examines the influence of one of the parameters of the Weibull distribution (m0) on the solution results. What can be said about the other parameters, how could they affect the results?

5. Figure 3(a) presents a comparison of the results obtained in this paper with the results of Ref. 29. How accurate is this comparison, since it seems that there is a different stress-strain state in these problems?

Reviewer 3 Report

The paper deals with the investigation of the mode I tensile opening failure of the alumina ceramic by using proposed model of a three dimensional hybrid finite - discrete element. The subject matter of this work seems to be appropriate to Modelling but the paper needs to be a bit improved.

Reviewer strongly suggests to add the Nomenclature.

Reviewer think that it will be more legible to put discussed on pages 8-9 data (above Eq. 9) in to tables.
